# Numerical Simulations of Sudden Oil Spills in Typical Cross-Border Rivers in the Yangtze River Delta Region

**Fei He [1], Qiuying Lai [1], Jie Ma [1] , Geng Wei [1,2] and Weixin Li [1,*]**

1   Nanjing Institute of Environmental Sciences, Ministry of Ecology and Environment, Nanjing 210042, China
2   College of Harbour, Coastal and Offshore Engineering, Hohai University, Nanjing 210024, China
*   Correspondence: lwxnies@126.com

**Abstract:** The Taipu River is an important cross-border river in the Yangtze River Delta region and a direct channel connecting Taihu Lake and Huangpu River. Along the main stream of the Taipu River are many sources of water, such as the Wujiang, Dingzha, and Liantang Rivers. Many boats traverse these rivers transporting a wide variety of goods, including large quantities of oil, chemicals, and other dangerous goods. In the event of accidents on these vessels, spilled cargo will directly threaten the drinking water safety of people in the region. Aiming at simulating and assessing the environmental risks of sudden oil spills in rivers in the Yangtze River Delta region, this paper established a two-dimensional oil spill model of the typical transboundary Taipu River based on the MIKE21 water environment numerical simulation software developed by the Danish Institute of Water Conservancy. The established model will improve emergency response and treatment plans as well as our understanding of river oil spill progressions.

**Keywords:** Yangtze River Delta region; transboundary rivers; oil spill model; numerical simulation





## 1. Introduction

The Yangtze River Delta area is a densely urbanized area and one of the most developed and fastest-developing areas in China [1–3]. With the rapid economic development, the demand for energy resources has risen sharply, especially for oil products. The shipping of oil, which is relatively low-cost due to the great volumes transported, has increased with demand, resulting in frequent sudden oil spills in recent years. This has become a major safety hazard for water environmental security in the Yangtze River Delta region [4,5].

The Taipu River is an important cross-border river in the Yangtze River Delta region, spanning the Jiangsu, Zhejiang, and Shanghai provinces and cities [6,7]. There are many water sources along the main stream of the Taipu River, so once an oil spill occurs it will inevitably lead to transboundary pollution and threaten the drinking water safety of people across a wide region [8,9]. In view of this, it is of great significance to scientifically predict the spread of pollution after sudden oil spills in the transboundary region of the Yangtze River Delta. Furthermore, it is important to clarify the main factors controlling different types of events and assess the degrees of pollution, scope of impact, and risk levels associated with various incidents. These efforts will facilitate the establishment of effective and sensitive early warning systems and the formulation of response plans for sudden pollution incidents.

A lot of work has been conducted internationally on the numerical simulation of oil spills, including the OILMAP system in the United States [10] and the OSCAR system in Norway [11]. Domestic scholars have also conducted a lot of work on oil spill numerical simulation. Xu et al. have incorporated the HSY algorithm into the traditional Lagrangian oil spill model and established an oil spill prediction model based on uncertainty analysis [12]; Yu et al. successively established oil spill prediction models in Zhoushan waters [13]; and Li et al. used hydrodynamic current field data and wind field data to simulate the influence

range of oil using a multi-module coupling oil spill model [14]. The above work is basically focused on the numerical simulation of oil spills in the ocean or estuaries, and there is little research on the numerical simulation of river oil spills at home and abroad. Therefore, this paper aims to make up for the shortage of domestic oil spill simulations by the numerical simulation of the Taipu River oil spill accident. It also provides emergency plans for oil spill accidents.

The objective of this paper is to establish the ability to quickly predict the extent and degree of oil pollution and the impact on water source inlet after an oil spill accident. At the same time, it has the function of water quality forecasting and risk warning. Thus, a scientific and efficient early warning and emergency system for sudden oil spill accidents can be built. It provides a decision-making basis for handling the sudden oil spill accident of the Taipu River and technical support for the safety of water supply quality of water sources.

Taking the typical transboundary Taipu River in the Yangtze River Delta region as the research object, this paper established a two-dimensional oil spill model in the MIKE21 software environment created by the Danish Institute of Water Conservancy [15,16]. This model can facilitate risk assessment research for emergency management and provide a theoretical basis and technical support for the prevention, monitoring, early warning, emergency response, and follow-up management of oil spill events. Therefore, this research has high theoretical research value as a unique example of a river oil spill simulation and high practical application value as a tool to be used during Taipu River oil spill events.

## 2. Materials and Methods

### 2.1. Study Area

The Taipu River starts from the east bank of Taihu Lake in Miaogang Town of Wujiang City, Jiangsu Province and flows east to Chijiagang Village of Jinze Town, Qingpu County, in the territory of Shanghai; it connects the Huangpu River with the Xiao River at Nandagang of Liantang Town. Altogether, the river spans the Jiangsu, Zhejiang, and Shanghai provinces (cities), with a total length of 57.6 km, and is a typical cross-border river [6]. The portion in Wujiang City, Jiangsu Province is 40.5 km long, the portion in Jiashan County in Zhejiang Province is a lake 1.46 km in length, and the portion in Shanghai is 15.24 km long. The Taipu River is an important multifunctional channel in the Taihu Lake Basin. The Taipu River also has flood discharge, water supply, shipping, landscape, ecological, and other functions. It is an important part of the pilot area of the Taihu Lake Basin co-governance and shared ecological protection and the construction of a "clean water Green Corridor" proposed in the eco-green integration development strategy of the Yangtze River Delta. The Taipu River is also a downstream Shanghai Jinze drinking water source. Because the Taipu River is located in the river network area of the Taihu Lake Basin, the water system is dense and the surface runoff is large, so the oil spill accident will have a great impact on the surrounding area. According to meteorological observation statistics, the annual average dominant wind direction in this area is ESE~ENE, and the annual average wind speed is 2.8 m/s [17]. Following calculation requirements to ensure reliable results, such as the completeness of hydrological data and the determinism of model boundaries, the calculation area in this paper was limited to the Taipu River section from the Taipu Lock to the Lu River, with a total length of about 57.6 km, as shown in Figure 1.

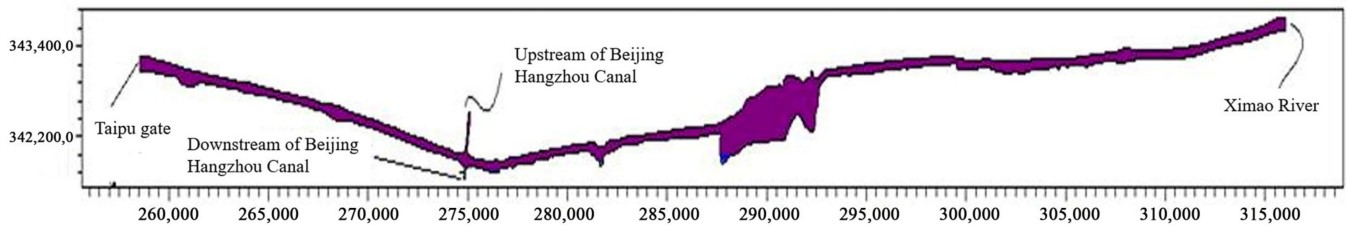

**Figure 1.** Typical cross-border river in the Yangtze River Delta region. Schematic diagram of the research area of the Taipu River (the dimensions in the picture are in meters).

### 2.2. Oil Spill Behavior and Calculation Principles

### 2.2.1. Transfer Process

The transport of oil particles involves both expansion and drift processes, and the composition of oil particles does not affect these processes [18].

Expansion movement: Oil film expansion is calculated using the modified Fay gravity–viscosity formula:

$$\left(\frac{dA_{oil}}{dt}\right) = K_a \cdot A_{oil}^{1/3} \cdot \left(\frac{V_{oil}}{A_{oil}}\right)^{4/3} \tag{1}$$

where $A_{oil}$ is the oil film area, $A_{oil} = \pi R_{oil}^2$; $R_{oil}$ is the oil film diameter; $K_a$ is the coefficient; and t is time.

Drift motion: the forces driving oil particle drift are water flow and wind pulling forces, and the total drift speed of oil particles can be calculated as:

$$U_{tot} = c_w(z) \cdot U_w + U_s \tag{2}$$

where $U_w$ is the wind speed at 10 m above the water surface; $U_s$ is the surface velocity; and $c_w$ is the wind drift coefficient, generally 0.03~0.04.

Turbulence spreading: assuming that horizontal diffusion is isotropic, the possible diffusion distance $S_\alpha$ in the $\alpha$ direction within a time step can be expressed as:

$$S_\alpha = [R]_{-1}^1 \cdot \sqrt{6 \cdot D_\alpha \cdot \Delta t_p} \tag{3}$$

where $[R]_{-1}^1$ is a random number from −1 to 1; $\Delta t_p$ is the diffusion time; and $D_\alpha$ is the diffusion coefficient in the $\alpha$ direction.

### 2.2.2. Weathering Process

Weathering of oil particles involves processes such as evaporation, dissolution, and emulsion formation, during which the composition of oil particles changes, but the horizontal position of the oil particles does not change [19].

Evaporation: Oil film evaporation is affected by oil content, air temperature, water temperature, oil spill area, wind speed, solar radiation, and oil film thickness. It is assumed that the diffusion in the oil film is unlimited, the temperature is higher than 0 °C, the thickness of the oil film is less than 5–10 cm, the oil film is completely mixed, and the partial pressures of some atmospheric components are negligible compared to the vapor pressure.

The evaporation rate can be expressed by the following equation:

$$N_i^e = k_{ei} \cdot P_i^{SAT} / RT \cdot \frac{M_i}{\rho_i} \cdot X \cdot \left[m^3/m^2s\right] \tag{4}$$

where N is the evaporation rate; $k_e$ is the material transport coefficient; $P^{SAT}$ is the vapor pressure; R is the gas constant; T is the temperature; M is the molecular weight; $\rho$ is the oil component density; and i is the oil component of interest. $K_{ei}$ is estimated from the following equation:

$$k_{ei} = k \cdot A_{oil}^{0.045} \cdot Sc_i^{-2/3} \cdot U_w^{0.78} \tag{5}$$

where k is the evaporation coefficient and $Sc_i$ is the Schmidt number for the vapor of component i.

Emulsification: Emulsification of oil spills refers to the process of mixing oil and seawater to form oil-water emulsions during weathering. Emulsification begins within hours of an oil spill. It depends on factors such as the thickness of the oil film, the density and viscosity of the spill itself, and the size of the wind waves [20].

(i)     Formation Process of Oil-In-Water Emulsions

The mechanisms driving oil into water include dissolution, diffusion, precipitation, etc., among which diffusion is the most important during the first few weeks after an oil spill. Diffusion is a mechanical process in which turbulent energy in the stream tears the oil film into oil droplets, forming an oil-in-water emulsion. These emulsions can be stabilized by surfactants that prevent the oil droplets from returning to the oil film. In bad weather, the dominant diffusion force comes from wave breaking, while in calm weather it comes from the stretching and compression of the oil film. The calculation of oil loss from oil film diffusion to the water body is as follows:

$$D = D_a \cdot D_b \tag{6}$$

where $D_a$ is the component entering the water body and $D_b$ is the component that does not return after entering the water body:

$$D_a = \frac{0.11(1 + U_w)^2}{3600} \tag{7}$$

$$D_b = \frac{1}{1 + 50\mu_{oil} \cdot h_s \cdot \gamma_{ow}} \tag{8}$$

where $\mu_{oil}$ is the viscosity of the oil and $\gamma_{ow}$ is the oil-water interfacial tension.

The rate at which the oil droplets return to the oil film is:

$$\frac{dV_{oil}}{dt} = D_a \cdot (1 - D_b) \tag{9}$$

(ii)    Formation Process of Water-In-Oil Emulsions

The changes of water content in oil can be expressed by the following equilibrium equation:

$$\frac{dy_w}{dt} = R_1 - R_2 \tag{10}$$

where $R_1$ and $R_2$ are the absorption rate and release rate of water, respectively, as calculated in the following equations:

$$R_1 = K_1 \cdot \frac{(1 + U_w)^2}{\mu_{oil}} \cdot (y_w^{max} - y_w) \tag{11}$$

$$R_2 = K_2 \cdot \frac{1}{As \cdot Wax \cdot \mu_{oil}} \cdot y_w \tag{12}$$

where $y_w^{max}$ is the maximum moisture content; $y_w$ is the actual water content; As is the bitumen content (weight ratio) in the oil; Wax is the content of paraffin in the oil (weight ratio); and $K_1$, $K_2$ are the absorption and release coefficients, respectively.

(iii)   Dissolution

Oil has a very weak tendency to dissolve into water, and its dissolving has little influence on the material balance calculation in the oil spill dynamic simulation, so it can be ignored in most cases [10]. However, in light of the ecological and social security significance of simulating the dissolution process of oil spills and predicting oil concentrations in water, the dissolution amount is generally calculated in oil spill models. The dissolved oil in water usually peaks within 8 to 12 h of the accident, and then the amount of dissolved oil decreases exponentially.

The dissolution rate Is expressed by the following equation:

$$\frac{dV_{ds_i}}{dt} = Ks_i \cdot C_i^{sat} \cdot X_{mol_i} \cdot \frac{M_i}{\rho_i} \cdot A_{oil} \tag{13}$$

where $C_i^{sat}$ is the solubility of component i; $X_{mol_i}$ is the mole fraction of component i; $M_i$ is the molar weight of component i; and $Ks_i$ is the dissolution mass transfer coefficient, which can be estimated from the following equation:

$$Ks_i = 2.36 \cdot 10^{-6} e_i \tag{14}$$

$$e_i = \begin{cases} 1.4 \text{ alkane} \\ 2.2 \text{ aromatic hydrocarbons} \\ 1.8 \text{ refined oils} \end{cases} \tag{15}$$

### 2.2.3. Heat Migration

Vapor pressure and viscosity are affected by temperature, and it has been observed that the temperatures of oil films are usually higher than those of the surrounding atmosphere and water (Figure 2).

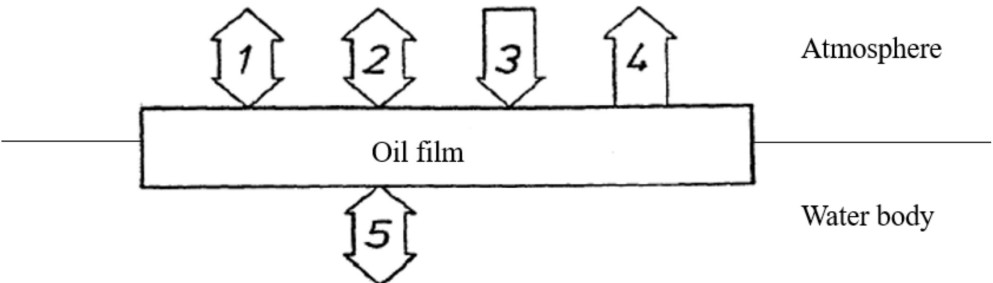

**Figure 2.** Schematic diagram of the heat balance of an oil film. 1 = heat transfer between the atmosphere and the oil film; 2 = heat radiation between atmosphere and oil film; 3 = solar radiation; 4 = evaporative heat loss; 5 = heat transfer between oil film and water.

### 2.3. Oil Spill Model Establishment

According to the characteristics of tide-affected rivers, a two-dimensional hydrodynamic model of the Taipu River was established using the FM module of the MIKE21 software package, which provided the hydrodynamic basis for the two-dimensional oil spill model. In combination with the oil particle calculation model, a two-dimensional oil spill model of the Taipu River was established with the help of the SA module in the MIKE21 software package, which can simulate the dynamic transport processes of oil spills, such as expansion, drift, and diffusion, under the joint actions of wind and hydraulic forces after the oil enters the water body. The simulation can project various oil spill parameters, such as oil film drift trajectory, time to reach sensitive waters, and attribute changes in oil products, which will assist in emergency decision making and damage assessment of oil spill accidents. Figure 3 is the flow chart of oil spill model construction in this paper.

The bed level of the river was determined by combining the relevant mapping data provided by the hydrological department and field-measured data from May 2015. According to the measured data from March 22 to March 24, 2015, the hydrological conditions of the Taipu River reach were calculated. The actual flow and water level of the Taipu River in the same period were taken as the upper and downstream boundary conditions, and the change process of the upper and downstream boundary hydrological elements of the two-dimensional hydrodynamic simulation in the study area was obtained through the simulation of a one-dimensional hydrodynamic mathematical model. The changes were taken as the boundary conditions of tidal flow volume and tidal level to simulate the hydrodynamic characteristics of the tidal flow process.

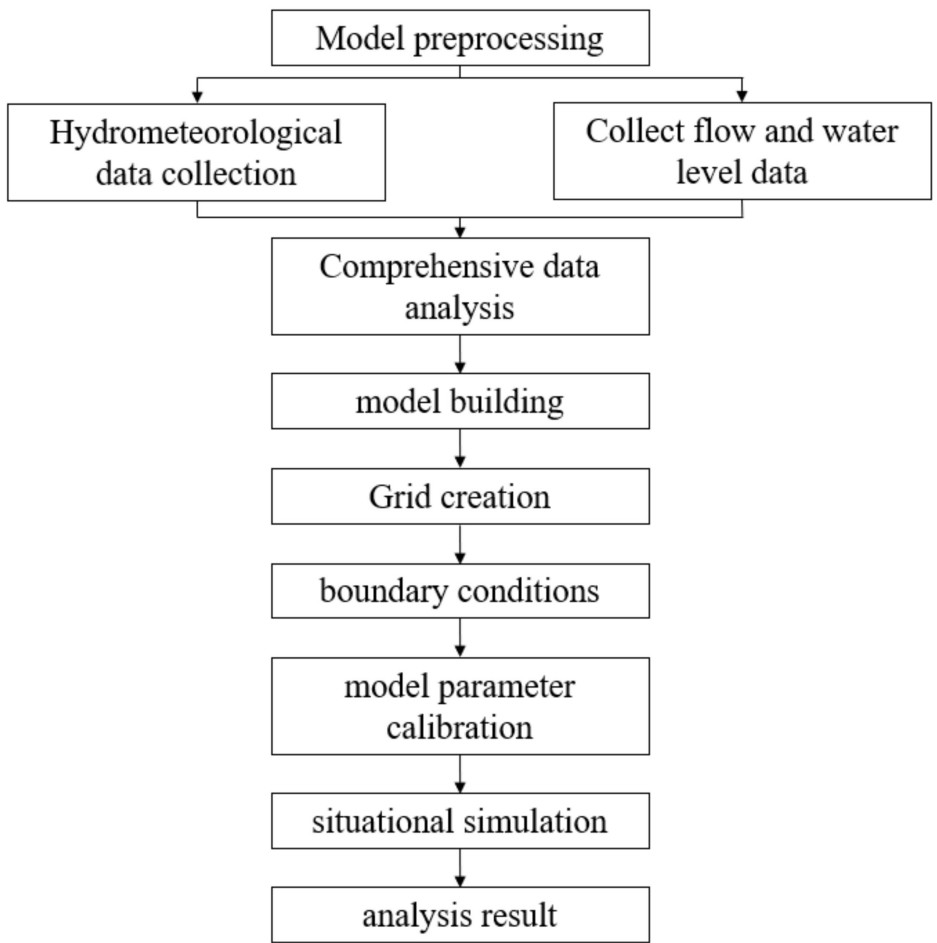

**Figure 3.** Flow chart of Taipu River oil spill model construction.

## 3. Results

### 3.1. Comparison of Calculated and Measured Water Level Values

The location of the oil film is mainly affected by the flow velocity and wind speed. Accurate simulation results of the flow field can ensure the accuracy of the simulation of the oil spill pollution accident to a certain extent. According to the hydrologic conditions determined by each boundary, the parameters of the established hydrodynamic model are calibrated. The value range of channel roughness is 0.018~0.022. For the established two-dimensional hydrodynamic model of the Taipu River, this study selects the measured cross-section data of Taipuzha, Pingwang Bridge, and Liantang Bridge in March to verify the water level of the model (Figure 4). The simulated and measured values were verified, and the relative errors were used to evaluate the applicability of the model and the accuracy of the simulation results. The relative error of the simulation results is less than 0.25, which is within the acceptable range. The results show that it is feasible to simulate the two-dimensional hydrodynamic model of the typical reaches of the Yangtze River, and the results can be used to simulate the oil spill pollution accident.

### 3.2. Oil Spill Accident Calculation and Analysis

Based on the reliability of the hydrodynamic simulation results, an oil spill approximating the Zhaojiabang oil spill accident in the upper reaches of the Taipu River was simulated and predicted according to the above-defined expansion, transport, and weathering processes. The calculated source intensity and related parameters of the simulated oil spill accident are shown in Table 1 and Figure 5.

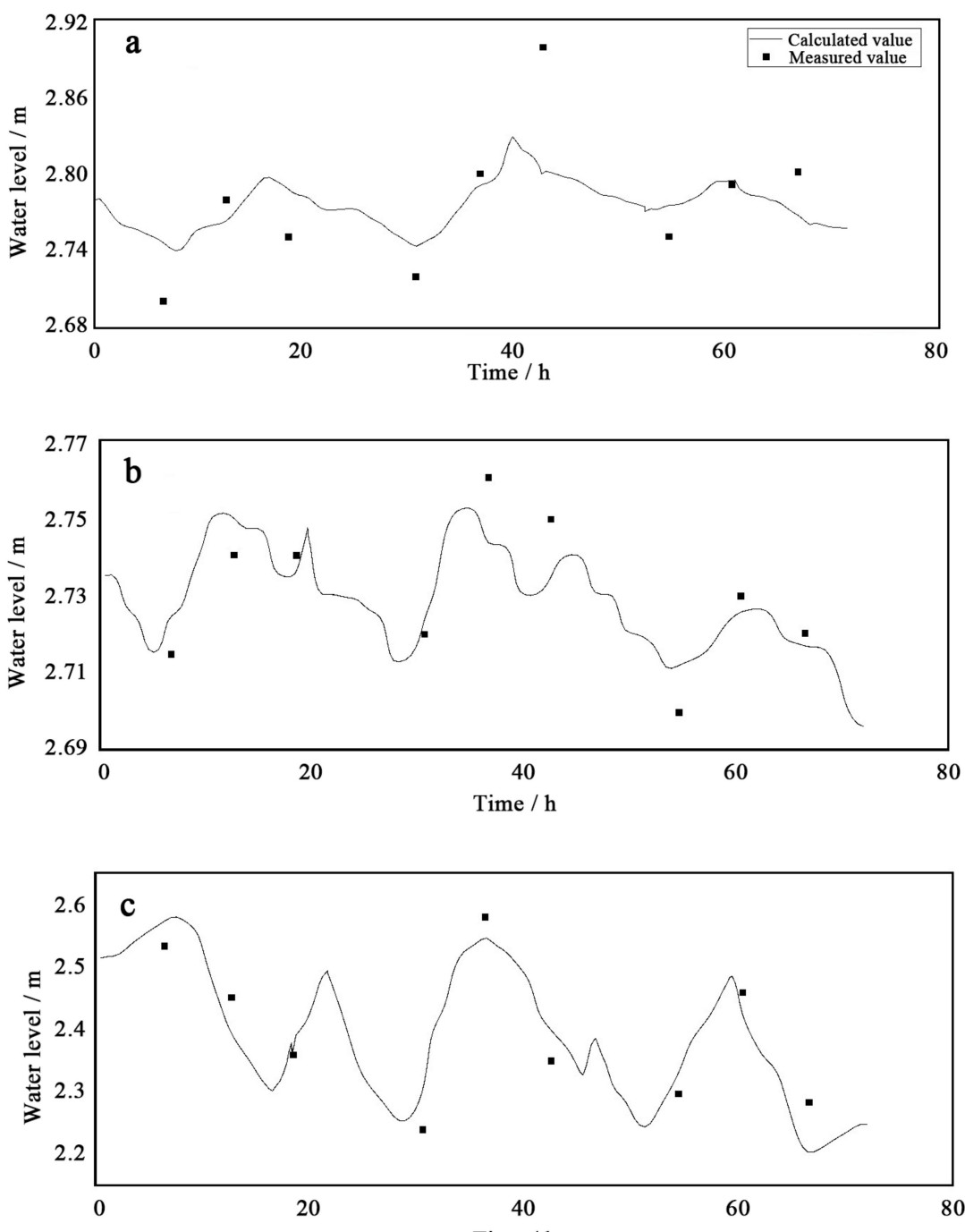

**Figure 4.** Comparison of calculated and measured water levels in each section in March ((**a**). Taipu Gate; (**b**). Pingwang Bridge; (**c**). Liantang Bridge).

**Table 1.** Calculated source strength and parameters related to the oil spill accident.

| Accident Parameter | Design Information Record |
|---|---|
| Spill location | Zhaojiabang in the upper reaches of the Taipu River |
| Leakage time | 30 min |
| Oil type | Light diesel oil |
| Quantity of oil spill | 30 t |
| Calculation step length | 15 s |

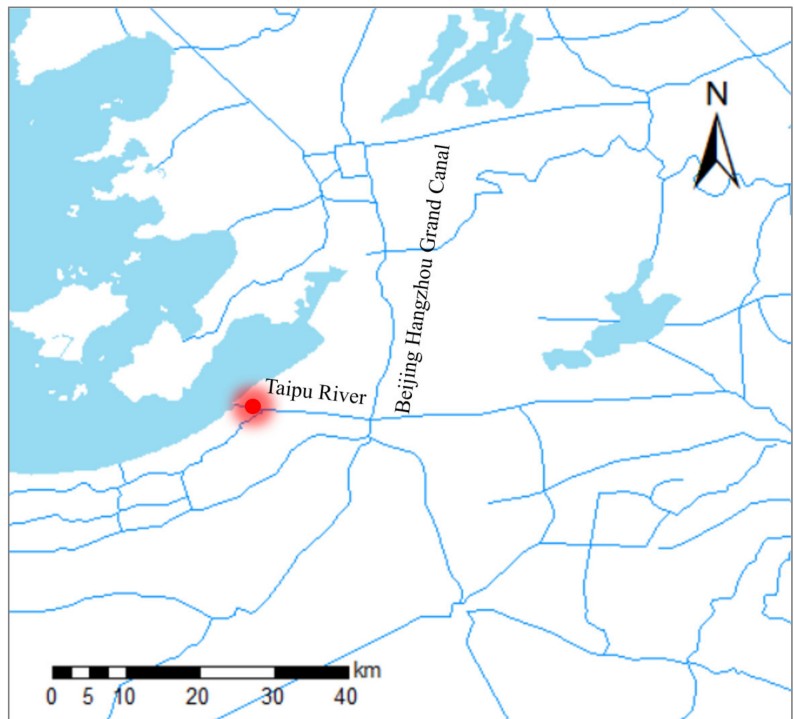

**Figure 5.** Map of the oil spill location in the upper reaches of the Taipu River.

The simulated oil spill indicated that the maximum thickness of the oil film decreased from 1.94 mm to 0.48 mm from the location of the spill to the end of the Taipu River. The diffusion and drift processes of the oil film in the Taipu River are shown in Figure 6.

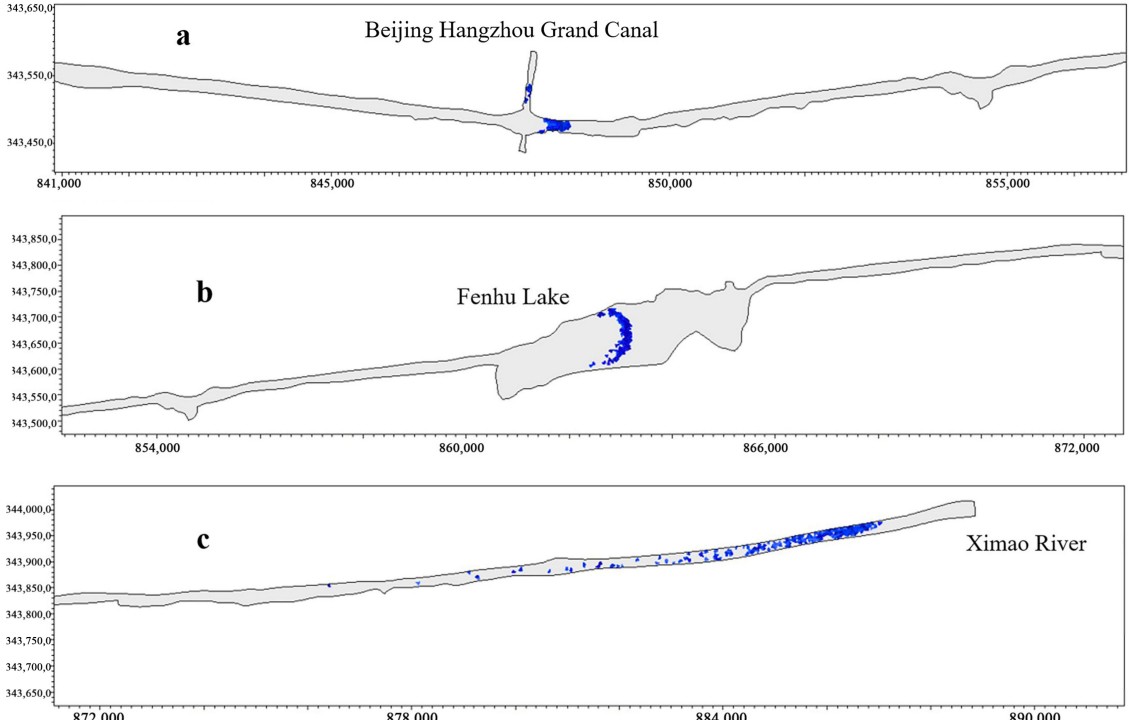

**Figure 6.** Simulation results of an oil spill accident in Taipu River (The blue dots represent the oil film distribution. (**a**). Section of Beijing Hangzhou Grand Canal; (**b**). section of Fenhu Lake; (**c**). section of Ximao River).

According to the calculation results, under the joint action of tidal currents and wind, the oil film formed by the oil spill generally drifts downstream in the direction of the coast. The above three local diagrams roughly illustrate the accident progression. The time taken by the oil film to reach the Beijing-Hangzhou Canal was 12 h, where the maximum thickness of the oil film was 1.05 mm; the time to reach the center of Fen Lake was 26 h, where the maximum local thickness was 0.64 mm; and the time to reach the lower boundary of the Taipu River was 48 h, where the maximum thickness was 0.48 mm. Oil spill evaporation mainly occurs within the first 10 h after an oil spill. At 0.2 h after the oil spill occurred, the oil spill evaporation rate was about 0.06%. In the 10th hour, the evaporation rate of oil spill was about 0.11. Subsequently, the evaporation rate of the oil spill slowed down greatly, and the evaporation rate remained at about 13%. An oil spill emulsifies under the action of turbulence and wind, and the combination of water and oil increases the water content of the oil spill, but the water content does not exceed 80%.

## 4. Discussions

This paper calculates the time and thickness of the oil film in different rivers by constructing an oil spill model of the Taipu River. The results show that the decline rate of oil spill thickness decreases with time. The evaporation rate and water content of the oil spill obtained by the model are close to those studied by other scholars [21,22], and the model can be used to simulate oil spills in the Taipu River area. The oil spill model established in this paper can well-simulate the migration and transformation processes of oil film expansion, oil particle transport with the flow, wind conduction drift, weathering and emulsification after an oil spill. It provides data support for research and judgment of oil spill accident risk situations and a scientific basis for taking effective emergency measures in time. However, the diffusion-drift of oil spill film is realized under the joint drive of hydraulic power and wind power, and its complexity determines that the dynamic simulation of oil spill behavior is complicated and difficult [4,23,24]. In addition to the limitation of research period and conditions, this paper still has some shortcomings. In the oil spill model in this study, the surface flow velocity required by hydrodynamic conditions was calculated by the vertical average flow velocity model calculated by the FM module through the built-in empirical formula, and was not checked stratified according to the measured vertical flow velocity. In order to obtain more accurate surface velocity, we should pay more attention to the stratification law of vertical velocity. In the oil spill model, the key parameters describing the oil spill processes of different oil products, such as extension, adsorption, and emulsification, need to be further verified by experiments.

## 5. Conclusions

According to the environmental risk characteristics of a sudden oil spill accident in the Taihu Basin, the Taipu River, a typical transboundary river, is the main research object in this paper. Based on the MIKE21 water environment numerical simulation software package developed by the Danish Institute of Water Environment, a two-dimensional oil spill model of the Taipu River was established. Through model calibration and verification, it is proved that the model has met the requirement of early warning accuracy. The arrival time and thickness of oil film in different rivers after an oil spill accident are simulated from the angles of oil film length, drift speed, and influence range of upstream and downstream. The main contents include the following:

(1) For the two-dimensional hydrodynamic model of the Taipu River, the measured data of three sections in March were selected to verify the water level of the model. The relative error of the simulation results was less than 0.25, which was within the acceptable range. It shows that the model is feasible to simulate the two-dimensional hydrodynamic dynamics of the typical reaches of the Yangtze River, and the model can be used to simulate the oil spill accident in the Taipu River area.

(2) Through the simulation of the oil spill accident at Zhaojiabang in the upper reaches of the Taipu River, the results show that the evaporation rate and water content of the oil

spill are close to the results of other scholars. The decreasing rate of oil film thickness gradually decreases with time, and the maximum thickness of oil film decreases from 1.94 mm to 0.48 mm.

(3)　The established oil spill model can better simulate the migration and transformation processes such as oil film expansion, oil particle transport with the flow, wind conduction drift, weathering, and emulsification after oil spill. The establishment of this model can reduce the blindness of the emergency treatment of oil spill accidents, and is of great significance to the establishment of risk assessment and decision management systems for oil spill accidents in the Taipu River.

**Author Contributions:** Conceptualization, F.H. and Q.L.; data curation, G.W.; formal analysis, Q.L.; investigation, F.H. and J.M.; methodology, F.H.; supervision, W.L.; visualization, J.M.; writing—original draft, Q.L.; writing—review & editing, F.H. All authors have read and agreed to the published version of the manuscript.

**Funding:** This research was funded by The Project of Ecological and Environmental Protection Integration Research Institute in Yangtze River Delta (No. ZX2022QT046), Major Science and Technology Program for Water Pollution Control and Treatment (No. 2017ZX07301006), and The Special Fund of Chinese Central Government for Basic Scientific Research Operations in Commonweal Research Institute (No. GYZX220405).

**Institutional Review Board Statement:** Not applicable.

**Informed Consent Statement:** Not applicable.

**Data Availability Statement:** The data used during the study appear in the submitted article.

**Conflicts of Interest:** The authors declare no conflict of interest. The funders had no role in the design of the study; in the collection, analyses, or interpretation of data; in the writing of the manuscript, or in the decision to publish the results.

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
