# Peer review of "Numerical Simulations of Sudden Oil Spills in Typical Cross-Border Rivers in the Yangtze River Delta Region"

_applsci, doi:10.3390/app122413029_

Round 1
Reviewer 1 Report
I have read with interest your paper on Numerical simulations of sudden oil spills in typical cross-border rivers in the Yangtze River Delta region. The manuscript deals with the interesting problem, however, I have a few major as well as minor comments.
Article presents results of numerical simulations of oil spills in rivers. The scope of investigation, lack of methodical discussion and wide result analysis related to other investigation and environmental determinants classified this manuscript as a technical report. Moreover, some parts of the text should be seriously improved. Introductory literature review is very poor. It should be widened, especially in context of aim of the study and conclusions. Concluding that ‘…model can accurately describe…’ or ‘…model can effectively simulate…’ is not proved by presentation of data results and comparison with other publications (investigations).
The applied methods are not described in procedural and interpretative matter. There is no results data presentation and its interpretation. We can only watch some pictures but question of spatial and temporal intensity of spill migration remains undiscovered.
In consequence of manuscript improvement, the ‘conclusions’ section has to be reconstructed.
Minor comments:
l. 20: keywords should contain some terms with modelling
Fig. 2. Mark atmosphere and water body in diagram. If possible, give the symbols (not numbers) of the processes for increased readability.
l. 200-205 ‘Good’ agreement between observed and estimated water levels is a very subjective statement. In order to assess this question more objective, we need to analyse statistical parameters of this comparison: correlation, determination, statistical significance, etc.
Fig. 3. The main title of the graph and names of gauging stations in particular boxes will be enough.
Author Response
Dear Editor and Reviewers:
Thank you for your letter and for the reviewers’ comments concerning our manuscript entitled “Numerical simulations of sudden oil spills in typical cross-border rivers in the Yangtze River Delta region”. Those comments are all valuable and very helpful for revising and improving our paper, as well as the important guiding significance to our research. Taking account of reviewers’ comments, we have revised and improved the manuscript. We hope our revisions meet with approval. Revised portion is marked up using the “Track Changes” function in the paper. The main corrections in the paper and the responses to the reviewers’ comments are as follows.
Reviewer 1
Comments and Suggestions for Authors
I have read with interest your paper on Numerical simulations of sudden oil spills in typical cross-border rivers in the Yangtze River Delta region. The manuscript deals with the interesting problem, however, I have a few major as well as minor comments.
Article presents results of numerical simulations of oil spills in rivers. The scope of investigation, lack of methodical discussion and wide result analysis related to other investigation and environmental determinants classified this manuscript as a technical report. Moreover, some parts of the text should be seriously improved. Introductory literature review is very poor. It should be widened, especially in context of aim of the study and conclusions. Concluding that ‘…model can accurately describe…’ or ‘…model can effectively simulate…’ is not proved by presentation of data results and comparison with other publications (investigations).
The applied methods are not described in procedural and interpretative matter. There is no results data presentation and its interpretation. We can only watch some pictures but question of spatial and temporal intensity of spill migration remains undiscovered.
In consequence of manuscript improvement, the ‘conclusions’ section has to be reconstructed.
Minor comments:
- 20: keywords should contain some terms with modelling
Fig. 2. Mark atmosphere and water body in diagram. If possible, give the symbols (not numbers) of the processes for increased readability.
- 200-205 ‘Good’ agreement between observed and estimated water levels is a very subjective statement. In order to assess this question more objective, we need to analyse statistical parameters of this comparison: correlation, determination, statistical significance, etc.
Fig. 3. The main title of the graph and names of gauging stations in particular boxes will be enough.
Point 1: The scope of investigation, lack of methodical discussion and wide result analysis related to other investigation and environmental determinants classified this manuscript as a technical report.
Response 1: Thanks for the reviewer’s suggestion. We have increased more statement on page 2, line 78-86, page 10, line 286-306 of the revised version.
(1) Taipu River is an important multifunctional channel in Taihu Lake Basin. Taipu Riv-er also has flood discharge, water supply, shipping, landscape, ecology and other func-tions. It is an important part of the pilot area of the Taihu Lake Basin co-governance and shared ecological protection and the construction of "clean water Green Corridor" proposed in the eco-green integration development strategy of the Yangtze River Delta. Taipu River is also downstream Shanghai Jinze drinking water source water source. Because Taipu River is located in the river network area of Taihu Lake Basin, the water system is dense and the surface runoff is large, the oil spill accident will have a great impact on the surrounding area.
(2) 4 Discussions
This paper calculates the time and thickness of the oil film in different rivers by constructing the oil spill model of Taipu River. The results show that the decline rate of oil spill thickness decreases with time. The evaporation rate and water content of oil spill obtained by the model are close to those studied by other scholars, and the model can be used to simulate oil spill in Taipu River area. The oil spill model established in this paper can well simulate the migration and transformation processes of oil film expansion, oil particle transport with the flow, wind conduction drift, weathering and emulsification after oil spill. It provides data support for research and judgment of oil spill accident risk situation and scientific basis for taking effective emergency measures in time. However, the diffusion-drift of oil spill film is realized under the joint drive of hydraulic power and wind power, and its complexity determines that the dynamic simulation of oil spill behavior is a complicated and difficult work. In addi-tion to the limitation of research period and conditions, this paper still has some shortcomings. In the oil spill model in this study, the surface flow velocity required by hydrodynamic conditions was calculated by the vertical average flow velocity model calculated by the FM module through the built-in empirical formula, and was not checked stratified according to the measured vertical flow velocity. In order to obtain more accurate surface velocity, we should pay more attention to the stratification law of vertical velocity. In the oil spill model, the key parameters describing the oil spill processes of different oil products, such as extension, adsorption and emulsification, need to be further verified by experiments.
Point 2: Moreover, some parts of the text should be seriously improved. Introductory literature review is very poor. It should be widened, especially in context of aim of the study and conclusions.
Response 2: Thanks for the reviewer’s suggestion. We have increased more statement on page 2, line 51-59 and rewritten the conclusions on page 10, line 308-345 of the revised version.
(1) Therefore, this paper aims to make up for the shortage of domestic oil spill simulation by numerical simulation of Taipu River oil spill accident. It also provides emergency plans for oil spill accidents.
The objective of this paper is to establish the ability to quickly predict the extent and degree of oil pollution and the impact on water source inlet after an oil spill acci-dent. At the same time, it has the function of water quality forecast and risk warning. Thus, a scientific and efficient early warning and emergency system for sudden oil spill accidents can be built. It provides decision-making basis for handling the sudden oil spill accident of Taipu River and technical support for the safety of water supply qual-ity of water source.
(2) 5 Conclusions
According to the environmental risk characteristics of sudden oil spill accident in Taihu Basin, Taipu River, a typical transboundary river, is the main research object in this paper. Based on the MIKE21 water environment numerical simulation software package developed by Danish Institute of Water Environment, a two-dimensional oil spill model of Taipu River was established. Through model calibration and verification, it is proved that the model has met the requirement of early warning accuracy. The ar-rival time and thickness of oil film in different rivers after oil spill accident are simu-lated from the Angle of oil film length, drift speed and influence range of upstream and downstream. The main contents include the following:
1) For the two-dimensional hydrodynamic model of the Taipu River, the meas-ured data of three sections in March were selected to verify the water level of the mod-el. The relative error of the simulation results was less than 0.25, which was within the acceptable range. It shows that the model is feasible to simulate the two-dimensional hydrodynamic dynamics of the typical reaches of the Yangtze River, and the model can be used to simulate the oil spill accident in the Taipu River area.
2) Through the simulation of the oil spill accident at Zhaojiabang in the upper reaches of Taipu River, the results show that the evaporation rate and water content of the oil spill are close to the results of other scholars. The decreasing rate of oil film thickness gradually decreases with time, and the maximum thickness of oil film de-creases from 1.94mm to 0.48 mm.
3) The established oil spill model can better simulate the migration and trans-formation processes such as oil film expansion, oil particle transport with the flow, wind conduction drift, weathering and emulsification after oil spill. The establishment of this model can reduce the blindness of the emergency treatment of oil spill accident and is of great significance to the establishment of risk assessment and decision man-agement system of oil spill accident in Taipu River.
Point 3: Concluding that ‘…model can accurately describe…’ or ‘…model can effectively simulate…’ is not proved by presentation of data results and comparison with other publications (investigations).
Response 3: Thanks for the reviewer’s suggestion. We have increased more statement on page 6, line 219-231 of the revised version.
The location of the oil film is mainly affected by the flow velocity and wind speed. Accurate simulation results of the flow field can ensure the accuracy of the simulation of oil spill pollution accident to a certain extent. According to the hydrologic condi-tions determined by each boundary, the parameters of the established hydrodynamic model are calibrated. The value range of channel roughness is 0.018~0.022. For the es-tablished two-dimensional hydrodynamic model of the Taipu River, this study selects the measured cross-section data of Taipuzha, Pingwang Bridge and Liantang Bridge in March to verify the water level of the model (Figure 4). The simulated and measured values were verified, and the relative errors were used to evaluate the applicability of the model and the accuracy of the simulation results. The relative error of simulation results is less than 0.25, which is within the acceptable range. The results show that it is feasible to simulate the two-dimensional hydrodynamic model of the typical reaches of the Yangtze River, and the results can be used to simulate the oil spill pollution accident.
Point 4: The applied methods are not described in procedural and interpretative matter. There is no results data presentation and its interpretation.
Response 4: Thanks for the reviewer’s suggestion. We have added a flow chart on page 6, line 205 to illustrate the construction process of the Taipu River oil spill model. We have described diagrams in detail, as detailed on page 3, line 94, page 7, line 241-247, page 9, line 262-267 of the revised version.
Point 5: We can only watch some pictures but question of spatial and temporal intensity of spill migration remains undiscovered.
Response 5: Thanks for the reviewer’s suggestion. We have increased more statement on page 10, line 274-280 of the revised version.
Oil spill evaporation mainly occurs within the first 10 hours after oil spill. At 0.2 hours after the oil spill occurred, the oil spill evaporation rate is about 0.06%. In the 10th hour, the evaporation rate of oil spill was about 0.11. Subsequently, the evaporation rate of oil spill slowed down greatly, and the evaporation rate remained at about 13%. Oil spill emulsifies under the action of turbulence and wind, and the combination of water and oil increases the water content of oil spill, but the water content does not exceed 80%.
Point 6: l. 20: keywords should contain some terms with modelling
Response 6: Thanks for the reviewer’s suggestion. We have added "Oil spill model; numerical simulation " on page 1, line 21 of the revised version.
Point 7: Fig. 2. Mark atmosphere and water body in diagram. If possible, give the symbols (not numbers) of the processes for increased readability.
Response 7: Thanks for the reviewer’s suggestion. We have added atmosphere and water body in diagram and modified the diagram on page 5, line 186 of the revised version.
Point 8: l. 200-205 ‘Good’ agreement between observed and estimated water levels is a very subjective statement. In order to assess this question more objective, we need to analyse statistical parameters of this comparison: correlation, determination, statistical significance, etc.
Response 8: Thanks for the reviewer’s suggestion. We have increased the analysis of the data between observed and estimated water levels on page 6, line 228-229 of the revised version.
Point 9: Fig. 3. The main title of the graph and names of gauging stations in particular boxes will be enough.
Response 9: Thanks for the reviewer’s suggestion. We have removed the excess from Figure 3 on page 7, line 241-246 of the revised version.

Reviewer 2 Report
The manuscript presents the results of studying “Numerical simulations of sudden oil spills in typical cross-border rivers in the Yangtze River Delta region”. This is important study for modeling the spread of oil pollution in watercourses and reservoirs. This manuscript has four sections including Introduction, Materials and methods, Results, and Conclusion. After reviewing this manuscript carefully, this manuscript was written very well. but there are some changing/suggestions required before publishing in worthy journal.
L 77, Fig. 1. Remove space in diagram word
L 96-97 Please, define ∆tp
Fig. 2. What is the difference between 5 and 6? Too large scale for such a simple scheme. As we can see the temperature difference from Fig. 2?
L 204. You need to evaluate the level of error in the calculations in comparison with the measurements in order to conclude that the convergence is good.
Fig. 3. There is no need to indicate in the caption of the figure what it shows. You cannot repeat three times that this is a comparison of calculated and measured water levels, but write: Comparison of calculated and measured water level at: a. Taipu Gate; b. …..
Fig. 5. What is shown in fig. 5: suspended or oil film thickness?
L 224. I do not see on the scales for the Fig. 5 the value of the oil film 1.94 mm. Does such a detailed gradation in color (oil film thickness) really correspond to the accuracy of the calculations? Error evaluation is required.
Please add to the discussion the significance of the findings. In particular, how will the danger of an oil film with a thickness of 1.019 mm differ from the danger of a film of 1.005 mm? What is the meaning of calculating and constructing the zoning of thicknesses that differ by a very small amount?
Conclusions should contain the results of research. It is not necessary to repeat the object of research and methods several times. Write specifically what and why was obtained as a result of modeling and how it can be applied when cleaning the river from oil.
Author Response
Dear Editor and Reviewers:
Thank you for your letter and for the reviewers’ comments concerning our manuscript entitled “Numerical simulations of sudden oil spills in typical cross-border rivers in the Yangtze River Delta region”. Those comments are all valuable and very helpful for revising and improving our paper, as well as the important guiding significance to our research. Taking account of reviewers’ comments, we have revised and improved the manuscript. We hope our revisions meet with approval. Revised portion is marked up using the “Track Changes” function in the paper. The main corrections in the paper and the responses to the reviewers’ comments are as follows.
Reviewer 2
Comments and Suggestions for Authors
The manuscript presents the results of studying “Numerical simulations of sudden oil spills in typical cross-border rivers in the Yangtze River Delta region”. This is important study for modeling the spread of oil pollution in watercourses and reservoirs. This manuscript has four sections including Introduction, Materials and methods, Results, and Conclusion. After reviewing this manuscript carefully, this manuscript was written very well. but there are some changing/suggestions required before publishing in worthy journal.
Point 1: L 77, Fig. 1. Remove space in diagram word
Response 1: Thanks for the reviewer’s suggestion. We have removed the Spaces on page 3, line 93 of the revised version.
Point 2: L 96-97 Please, define ∆tp
Response 2: Thanks for the reviewer’s suggestion. We have defined ∆tp on page 3, line 112 of the revised version.
Point 3: Fig. 2. What is the difference between 5 and 6? Too large scale for such a simple scheme. As we can see the temperature difference from Fig. 2?
Response 3: Thanks for the reviewer’s suggestion. We have made adjustments to Figure 2 on page 6, line 186 of the revised version. Because the oil film density is small floating on the sea surface and the oil film is airtight. As a result, the oil coating on the surface of the sea can prevent the evaporation of seawater, resulting in lower humidity and air drying in the polluted area and surrounding areas. The oil film will increase the reflectivity of the sea surface and reduce solar radiation into the water. This causes the water below the surface of the oil film to cool down. At the same time, the oil film can absorb solar radiation and increase the surface water temperature.
Point 4: L 204. You need to evaluate the level of error in the calculations in comparison with the measurements in order to conclude that the convergence is good.
Response 4: Thanks for the reviewer’s suggestion. We have increased more statement on page 6, line 228-229 of the revised version.
Point 5: Fig. 3. There is no need to indicate in the caption of the figure what it shows. You cannot repeat three times that this is a comparison of calculated and measured water levels, but write: Comparison of calculated and measured water level at: a. Taipu Gate; b. …..
Response 5: Thanks for the reviewer’s suggestion. We have changed the caption of the figure on page 7, line 241-246 of the revised version.
Point 6: Fig. 5. What is shown in fig. 5: suspended or oil film thickness?
Response 6: Thanks for the reviewer’s suggestion. We have explained the contents of fig. 5 in detail on page 9, line 265-267 of the revised version.
Point 7: L 224. I do not see on the scales for the Fig. 5 the value of the oil film 1.94 mm. Does such a detailed gradation in color (oil film thickness) really correspond to the accuracy of the calculations? Error evaluation is required.
Response 7: Thanks for the reviewer’s suggestion. 1.94mm is the maximum thickness of oil film in the whole simulation process. Fig. 5 shows the simulation results of oil spill distribution in the three regions after a period of oil spill, so 1.95mm is not contained. Models can achieve such detailed color gradients (oil film thickness). But this is not necessary, so we have deleted this part.
Point 8: Please add to the discussion the significance of the findings. In particular, how will the danger of an oil film with a thickness of 1.019 mm differ from the danger of a film of 1.005 mm? What is the meaning of calculating and constructing the zoning of thicknesses that differ by a very small amount?
Response 8: Thanks for the reviewer’s suggestion. We have added discussions section on page 10, lines 286-306 of the revised version. The oil film thickness distribution has been removed from Figure 5 because we are more focused on simulating the time it takes the oil film to reach a certain area and its distribution.
4 Discussions
This paper calculates the time and thickness of the oil film in different rivers by constructing the oil spill model of Taipu River. The results show that the decline rate of oil spill thickness decreases with time. The evaporation rate and water content of oil spill obtained by the model are close to those studied by other scholars, and the model can be used to simulate oil spill in Taipu River area. The oil spill model established in this paper can well simulate the migration and transformation processes of oil film expansion, oil particle transport with the flow, wind conduction drift, weathering and emulsification after oil spill. It provides data support for research and judgment of oil spill accident risk situation and scientific basis for taking effective emergency measures in time. However, the diffusion-drift of oil spill film is realized under the joint drive of hydraulic power and wind power, and its complexity determines that the dynamic simulation of oil spill behavior is a complicated and difficult work. In addition to the limitation of research period and conditions, this paper still has some shortcomings. In the oil spill model in this study, the surface flow velocity required by hydrodynamic conditions was calculated by the vertical average flow velocity model calculated by the FM module through the built-in empirical formula, and was not checked stratified according to the measured vertical flow velocity. In order to obtain more accurate surface velocity, we should pay more attention to the stratification law of vertical velocity. In the oil spill model, the key parameters describing the oil spill processes of different oil products, such as extension, adsorption and emulsification, need to be further verified by experiments.
Point 9: Conclusions should contain the results of research. It is not necessary to repeat the object of research and methods several times. Write specifically what and why was obtained as a result of modeling and how it can be applied when cleaning the river from oil.
Response 9: Thanks for the reviewer’s suggestion. We have rewritten conclusions of the article on page 10, lines 307-345 of the revised version.
5 Conclusions
According to the environmental risk characteristics of sudden oil spill accident in Taihu Basin, Taipu River, a typical transboundary river, is the main research object in this paper. Based on the MIKE21 water environment numerical simulation software package developed by Danish Institute of Water Environment, a two-dimensional oil spill model of Taipu River was established. Through model calibration and verification, it is proved that the model has met the requirement of early warning accuracy. The arrival time and thickness of oil film in different rivers after oil spill accident are simulated from the Angle of oil film length, drift speed and influence range of upstream and downstream. The main contents include the following:
(1) For the two-dimensional hydrodynamic model of the Taipu River, the measured data of three sections in March were selected to verify the water level of the model. The relative error of the simulation results was less than 0.25, which was within the acceptable range. It shows that the model is feasible to simulate the two-dimensional hydrodynamic dynamics of the typical reaches of the Yangtze River, and the model can be used to simulate the oil spill accident in the Taipu River area.
(2) Through the simulation of the oil spill accident at Zhaojiabang in the upper reaches of Taipu River, the results show that the evaporation rate and water content of the oil spill are close to the results of other scholars. The decreasing rate of oil film thickness gradually decreases with time, and the maximum thickness of oil film decreases from 1.94mm to 0.48 mm.
(3) The established oil spill model can better simulate the migration and transformation processes such as oil film expansion, oil particle transport with the flow, wind conduction drift, weathering and emulsification after oil spill. The establishment of this model can reduce the blindness of the emergency treatment of oil spill accident and is of great significance to the establishment of risk assessment and decision management system of oil spill accident in Taipu River.

Round 2
Reviewer 1 Report
Paper can be accepted on condition that it will be published as ‘technical note’ or similar category
Reviewer 2 Report
The authors have done a great job of improving the manuscript. I believe that in its current form the article can be accepted for publication.